# Increased CT Use and No Change in Injury Severity among Child Motor Vehicle Victims: A National Trauma Database Study in Japan

**DOI:** 10.3390/healthcare11091240

**Published:** 2023-04-26

**Authors:** Wataru Ishii, Masahito Hitosugi, Kenji Kandori, Michitaro Miyaguni, Ryoji Iizuka

**Affiliations:** 1Emergency of Medicine, Critical Care Center, Kyoto Daini Red Cross Hospital, Haruobi, Kamazamarutamachi, Kamigyo, Kyoto 602-8026, Japan; wataruaug0804@lily.ocn.ne.jp (W.I.); knj.kandori@gmail.com (K.K.); mitch.miyaguni10@gmail.com (M.M.); ryouji_iiduka@outlook.jp (R.I.); 2Department of Legal Medicine, Shiga University of Medical Science, Tsukinowa, Seta, Otsu 520-2192, Japan

**Keywords:** motor vehicle passenger, road traffic collision, child trauma, Japan Trauma Data Bank, computed tomography

## Abstract

The number of fatalities associated with traffic accidents has been declining owing to improvements in vehicle safety performance and changes in the law. However, injuries in children can lead to social and economic losses. We examined 10-year changes in the characteristics of traffic trauma among pediatric motor vehicle passengers by analyzing data from the Japan Trauma Data Bank (JTDB). Among the 36,715 injured motor vehicle passengers under the age of 15 years who were registered in the JTDB from 2004 to 2019, we compared the groups injured during 2004–2007 (*n* = 94) and 2017–2019 (*n* = 203). Physiologically, the 2004–2007 group had a lower body temperature and Glasgow Coma Scale score as well as a higher mortality. Anatomical severity was higher in the 2004–2007 group for the head, face, and neck, according to the Abbreviated Injury Scale. In terms of treatment, only craniotomy as a primary surgery was significantly lower in the 2017–2019 group. The 2017–2019 group had significantly higher rates of receiving whole-body computed tomography (CT). Because the rate of performing CT has increased, with no changes in the injury severities of the trunk and extremities, limiting the number of CT examinations is suggested for pediatric motor vehicle passengers involved in road traffic collisions. The severity of trunk and extremity injuries has not improved in more than 10 years; further preventive measures for these injuries should be considered.

## 1. Introduction

Injuries in children can lead to social and economic losses, and with severe trauma, children often have long-term disabilities [1,2]. Krzysztof et al. reported that in the child injury pyramid for each fatally injured child under the age of 19 years, 45 children required hospitalization and a further 1300 needed medical care at an outpatient emergency clinic [3]. Furthermore, approximately half of children under age 12 years who are treated in emergency departments have ongoing physical, mental, or sociological disabilities [3]. In recent years, the number of deaths associated with road traffic collisions (RTCs) has been decreasing owing to improvements in vehicle safety performance and changes in legislation. However, because injuries owing to RTCs are the leading cause of death among children and young adults aged 5–29 years globally, the prevention of pediatric deaths is an urgent issue in all countries. In Japan, the number of fatalities owing to traffic-accident-related trauma among people under 15 years of age decreased from 185 in 2004 to 47 in 2019. The number of casualties also decreased from 80,770 in 2004 to 26,589 in 2019 (Table 1). In terms of the manner of death and severe injuries, the rate of injury among preschool children who are motor vehicle passengers has been on the increase in recent years (10.6% in 1990 to 34.5% in 2019) [4]. To further increase the survival rate of pediatric motor vehicle passengers with severe injuries, it is important to understand the current characteristics and peculiarities of pediatric trauma to provide the same quality of primary trauma care as for adults. Children are not small adults; they have a different physical and anatomical structure, level of maturity, and interests. Therefore, understanding the characteristics of injuries in children is important to establish preventive measures. In this study, we examined the 10-year change in the characteristics of traffic trauma among pediatric motor vehicle passengers. The data obtained can be applied in primary trauma care for improvements in the quality of care and interventions for child motor vehicle passengers involved in RTCs.

## 2. Purpose

In this study, we analyzed data from the Japan Trauma Data Bank (JTDB) with the aim of providing valuable information regarding primary care for child motor vehicle passengers involved in RTCs, which can contribute to reducing the number, severity, and mortality of pediatric motor vehicle passengers. This study also contributes to improving the quality of medical interventions for injured children.

## 3. Materials and Methods

### 3.1. Study Design and Patient Selection

This observational study was a retrospective analysis of data from a national hospital-based database, the JTDB. The JTDB is a national trauma registry in Japan that includes data recorded by the Japanese Society of Trauma Surgeons and Japanese Society of Emergency Medicine. The database has been active since 2003. This registry is similar to trauma databases in North America, Europe, and Oceania [5,6,7]. Approximately 372,000 patients with trauma have been registered as of 2019 [8]. In 2005, 55 hospitals participated in the registry; however, in 2021, more than 303 hospitals participated, accounting for approximately 75% of all emergency centers in Japan. Although the number of registered hospitals has increased over the years, the proportion of emergency centers for the treatment of many severe traumas among them has not changed. The JTDB collects information on the mode and mechanism of trauma, vital signs, the anatomical and physiological severity of injury, pre-hospital and in-hospital treatments, and outcomes.

The data for the present study were obtained from the JTDB in December 2020. A total of 372,314 patients were registered in the JTDB from 2004 to 2019. Of these, 36,715 were injured motor vehicle passengers. We excluded the following motor vehicle passenger patients from 2008 to 2016: patients who arrived at the hospital with cardiopulmonary arrest; those whose age, accident year, or Abbreviated Injury Scale (AIS) score were unclear; and those aged more than 15 years. Finally, we obtained data for 297 motor vehicle passengers under the age of 15 years, with 94 patients from 2004 to 2007 and 203 from 2017 to 2019 (Figure 1). Because the JTDB has been active since 2003, the number of registered cases was small in the early years; therefore, to obtain a sufficient number of cases, we set a 4-year study period. However, because the number of registered cases has increased recently, we recently reset the study period to 3 years to yield greater equilibrium in the sample. Additionally, data through 2019 were analyzed because of changes in the AIS assessment from 2020 onward in the JTDB.

The following information was obtained for each patient: age, sex, seat position in the motor vehicle, vital signs upon hospital arrival (systolic and diastolic blood pressure, heart rate, respiratory rate, body temperature (BT), and Glasgow Coma Scale (GCS) score), focused assessment with sonography for trauma (FAST) test results, mortality rate, AIS scores for each body region (1998 version), Injury Severity Score (ISS), Revised Trauma Score (RTS), and Trauma and Injury Severity Score Probability of Survival (TRISSPS). The AIS score is used to classify the type and severity of injury in each body part on an anatomical scale of 1 (minor) to 6 (clinically untreatable). The ISS is used for assessing the severity of multiple injuries and is the sum of the squares of the highest AIS scores for each of the three most severely injured body parts. Because these scores correlate well with mortality, they are usually assessed as a measure of anatomic severity in an emergency medical setting; in Japan, scores are evaluated by medical staff who have participated in coding seminars. The RTS is a method of assessing severity based on physiological indices, with the highest score (least severe) being 7.84 and the lowest score (most severe) being 0. The TRISSPS sums the physiological and anatomical severity score of the patient’s condition with their age to calculate a predicted survival (Ps) rate as follows: (1) preventable death (Ps > 0.50), (2) possibly preventable death (0.25 < 0.50), and (3) non-preventable death (Ps < 0.25). Furthermore, medical intervention for each patient was also examined. Information was also obtained regarding performance of computerized tomography (CT) for each part, blood transfusion within 24 h of arrival, intervention radiology of each part, and primary surgery for each part.

The data in the JTDB are based on data from hospitals. The JTDB does not include data from the police department regarding the collision or road conditions. Owing to the lack of police crash data, we decided not to include patients who died at the scene of the accident in the analysis.

### 3.2. Statistical Analysis

Categorical variables are shown as proportion or frequency. Continuous variables are presented as mean ± standard deviation for values that follow a normal distribution and as median and interquartile range for values that are non-normally distributed. The Chi-square test was used to compare the prevalence between two groups. To identify differences in values between two groups, the Student’s *t*-test was used for values with a normal distribution, and the Mann–Whitney test was used for values with a non-normal distribution. A *p*-value of ≤0.05 was considered statistically significant. The analyses were performed with JMP Pro 15.2.0 (SAS Institute Inc., Cary, NC, USA).

### 3.3. Endpoint

We examined 10-year changes in characteristics of traffic-related trauma and in medical interventions among injured pediatric motor vehicle passengers.

## 4. Results

The obtained items were compared between the patients in the group of patients who were injured during 2004–2007 (*n* = 94) and the group injured during 2017–2019 (*n* = 203).

### 4.1. Comparison of Patient Characteristics and Information Obtained at Hospital Arrival

The BT and GCS scores were significantly lower in the 2004–2007 group than in 2017–2019 group (Table 2). When comparing the AIS scores in each body region, the 2004–2007 group had significantly higher AIS scores for the head, face, and neck than the 2017–2019 group (Table 3). Regarding injury severity, the RTS was significantly lower in the 2004–2007 group than that in the 2017–2019 group (Table 4).

### 4.2. Comparisons of Medical Interventions

In a comparison between the groups regarding the rates of receiving CT, there was no significant difference in the overall CT performance rate. However, the 2017–2019 group had significantly higher rates of receiving CT of the neck, chest, abdomen, pelvis, and spine. No significant difference was found with respect to transfusion rates within 24 h (Table 5). Next, the prevalence of medical interventions was examined between the two groups. No significant difference was found in the rate of intervention radiology between the 2004–2007 group and 2017–2019 group. Only craniotomy as a primary surgery was significantly lower in the 2017–2019 group. There was no significant difference between the two groups for other primary surgeries (Table 6).

## 5. Discussion

In the present study, AIS scores of the head were the highest among all body regions in both periods. Generally, children are more vulnerable to head injuries. One study examined Trauma Registry data for severe injuries in German pediatric motor vehicle passengers involved in a motor vehicle collision [9]. Although the included patients were aged 0 to 5 years and had cardiopulmonary arrest on arrival, different from our study, the results suggested that head injuries were most frequently observed, accounting for 56.0% of injuries [9]. Additionally, the prevalence of patients with a GCS score 8 or less increased significantly with decreases in age [9]. Among motor vehicle passengers, young children have the highest risk of head injuries.

Our results suggested that AIS scores of the head have decreased, GCS scores have increased, and the rate of craniotomy has decreased over the 10-year period of this study. These trends are similar to those obtained for adult motor vehicle passengers in Japan [10]. According to the study, patients’ GCS score and AIS scores of the head, face, abdomen, and lower extremities improved significantly over the most-recent 10-year period and subsequently, the survival rate decreased [10]. Factors affecting mortality among child motor vehicle passengers have also been examined. GCS scores (odds ratio (OR) 1.96), BT (OR 2.58) scores, and AIS scores of the head (OR 0.29) were identified as independent predictors of non-fatal outcomes [11]. A study using a trauma registry of children younger than age 15 years in New South Wales showed that a high ISS and an injured body region of the head or neck were predictors of in-hospital mortality [12]. A 10-year review of child injuries in Australia revealed that having a head injury and greater injury severity were associated with higher mortality rates [13]. In Japan, a report using the JTDB examined in-hospital mortality trends among injured patients between 2009 and 2018 [14]. The report suggested a significant decrease in mortality for the age group for the ages of 0–4 and 5–14 years [14]. Therefore, the trends among child vehicle passengers involved in RTCs in our study were in accordance with previous results that decreasing head injury severity and maintaining high GCS scores lead to reduced fatalities. Research using data of the Crash Injury Research Engineering Network in the United States examined factors influencing GCS scores in pediatric motor vehicle passengers. According to the study, a vehicle velocity in the crash (delta V) of more than 30 miles per hour significantly decreased the GCS scores [15]. Therefore, promoting road safety awareness and education for caregivers to reduce vehicle velocity is a priority.

There may be several reasons for the decreasing AIS values of the head, face, and neck and the increasing GCS over the 10-year study period. Although the use of child safety seats had been mandated, the use of seatbelts in rear-passenger seats was only mandated in the road traffic law in 2008. Therefore, we believe that more children who were rear-seat passengers used a seatbelt during 2017–2019. Safety improvements in motor vehicles and equipment made using precrash safety systems, as well as prehospital trauma care and widespread education in trauma care have also contributed to this change.

In this study, although injury severities of the trunk and extremities and the prevalence of treatment (except for craniotomy) did not change over the 10-year study period, the prevalence of performing CT for trunk and extremities increased significantly. This trend might be characteristic of Japan. Japan has the highest number of CT scanners per capita in the world [16]. In most emergency departments and critical care centers, high-speed CT scanners are located close to the trauma bay. Moreover, hybrid emergency rooms equipped with CT scanners have recently been installed in many facilities. Therefore, medical staff in Japan can easily perform CT for trauma patients. Furthermore, performing CT scans in early trauma care for patients with polytrauma has significantly improved outcomes and reduced mortality [17,18]. Furthermore, there is some evidence that the use of whole-body CT is associated with improved mortality for patients with trauma [19,20,21,22]. In primary surveys of severe trauma in Japan, FAST is performed for the early assessment of trunk injuries, followed by whole-body CT. Furthermore, many emergency physicians in Japan follow the Japan Advanced Trauma Evaluation and Care (JATEC) guidelines for primary trauma care, in which CT imaging is actively performed not only for pediatric but also adult trauma cases. According to the Pediatric Emergency Care Applied Research Network (PECARN) criteria, when multiple trauma injuries are suspected, performing CT is recommended, even for children [23]. Because this guideline has become increasingly popular, a larger number of CT imaging scans were performed during 2017–2019. According to a study using data of the Trauma Registry in Germany, FAST is performed in 87.1–94.0% and CT is performed for 74.3–80.0% of child passengers involved in a motor vehicle collision [9]. In our study, CT was administered in 86.1% of children. In another study dealing with child patients who had emergency CT for trauma, 32.4% received whole-body scans and 67.6% received a cranial scan [24]. Recently, an increased risk of cancer has been reported in the pediatric population owing to CT examinations [25]. According to the estimation by Frush et al., the risk of children developing a fatal cancer owing to radiation is approximately 1 in 1000 CT examinations [25]. An injured child may receive multiple scans on admission followed by repeat scanning to evaluate the progression of injuries during hospitalization. This can lead to a rapid accumulation of radiation doses, placing these children at increased risk for cancers [26]. Additionally, there has been greater reliance recently on the high diagnostic power of CT; however, few emergency physicians understand the risks associated with radiation [26,27,28]. There are a few solid decision-making tools to assess trauma in children. The PECARN criteria suggest that the CT indications for pediatric (age <18 years) head trauma include a GCS score of 14 or 15 and transport to the hospital within 24 h of injury [23]. Additionally, the Canadian Assessment of Tomography for Childhood Head Injury (CATCH) and Children’s Head Injury Algorithm for the Prediction of Important Clinical Events (CHALICE) are used in emergency departments. However, few reliable clinical alternatives to CT imaging for primary treatment are available [29,30]. Unnecessary CT use is an important problem not only for pediatric patients involved in motor vehicle collisions but also for adults [31]. However, suppressing CT in emergency medicine may increase diagnostic delays and the number of missed cases [32]. Reducing radiation exposure by curtailing unnecessary CT imaging is a priority; however, social institutional measures to protect emergency physicians from lawsuits are also needed. According to a recent study from Japan, use of whole-body CT did not reduce the in-hospital mortality of patients with trauma compared with the use of selective CT [33]. Because of children’s greater vulnerability owing to their size and developmental limitations, child vehicle passengers may experience more severe injuries when involved in a collision. CT has become an important adjunct to increasing the diagnostic acumen in the initial evaluation [34,35]. Therefore, CT is indispensable for injured children involved in motor vehicle collisions. However, as suggested by the concept “as low as reasonably achievable”, limiting the number of CT examinations is recommended for child motor vehicle passengers involved in RTCs.

Despite the decrease in mortality among injured children in Japan, motor vehicle-related injury is an important public health problem because injuries cause physical, psychological, and financial burdens for children, their families, and society. In this study, the severity of injuries to the trunk and extremities did not improve in more than 10 years; therefore, further preventive measures for these injuries should be considered. Furthermore, countries including Japan have committed to the Vision Zero framework to eliminate motor vehicle fatalities; thus, further development of vehicle safety systems is also required.

## 6. Conclusions

According to this nationwide trauma database study, the AIS scores of the head were highest among all body regions for child motor vehicle passengers involved in RTCs. When comparing patients injured during 2004–2007 and those injured during 2017–2019, the AIS scores of the head, face, and neck were significantly higher in the 2004–2007 group than those in the 2017–2019 group. Regarding injury severity, the RTS was significantly lower in the 2004–2007 group than that in the 2017–2019 group. The 2017–2019 group had significantly higher rates of receiving CT of the neck, chest, abdomen, pelvis, and spine. Craniotomy as a primary surgery was significantly lower in the 2017–2019 group than in the 2004–2007 group.

Because the rate of performing CT has increased, with no change in the injury severity of the trunk and extremities, limiting the number of CT examinations is recommended for child motor vehicle passengers involved in RTCs. Because the severity of injuries of the trunk and extremities has not improved for more than 10 years, further preventive measures for these injuries should be considered.

## 7. Limitation

There are several limitations in this study. First, this study did not cover all injured motor vehicle passengers in Japan. However, the JTDB includes as many trauma cases as databases in North America, Europe, and Oceania, because approximately 75% of emergency medical centers register data in this database. Thus, because the JTDB is the only nationwide hospital-based prospective trauma registry in Japan, the present analysis is expected to yield representative results. Second, crash details (type of motor vehicle, direction of impact, and speed) are not included in the JTDB registry. Future research should investigate the relationship between accident data collected by the police and medical data collected from Japanese hospital records. Third, there was an informational deficiency regarding the JTDB dataset. The JTDB does not allow the entry of textual information, although relevant items, such as numerical values, are registered through the website. Therefore, some items had missing data. However, the reliability of the analyzed data is high because unclear or missing data were excluded from the analysis. Fourth, we did not include deaths that occurred at the scene of a collision because this database is based on hospitals; we did not believe that including these out-of-hospital deaths would increase the reliability of the analysis. Fifth, in this study, patients were limited to motor vehicle passengers. Compared with motor vehicle passengers aged 0–16 years who are involved in RTCs, cyclists are 5.8 times, motorcyclists are 22.1 times, and pedestrians are 55 times more likely to be involved in a collision. Therefore, similar studies of other road users are warranted to prevent road-traffic-related fatalities among children.

## Figures and Tables

**Figure 1 healthcare-11-01240-f001:**
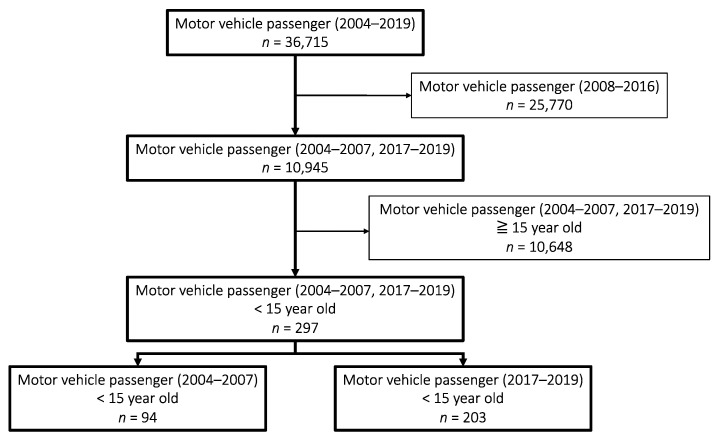
Flowchart of patient enrollment.

**Table 1 healthcare-11-01240-t001:** Changes in Pediatric Traffic Trauma Fatalities and Injuries in Japan.

	2004	2005	2006	2007	2017	2018	2019
Death cases	185	154	134	115	57	69	47
Casualty cases	80,770	78,654	72,752	70,098	34,563	30,208	26,589

**Table 2 healthcare-11-01240-t002:** Comparisons of patient characteristics and physiological parameters for the injury group with hospital arrivals from 2004 to 2007 and that from 2017 to 2019. FAST: focused assessment with sonography for trauma.

	2004 to 2007 Injury Group	2017 to 2019 Injury Group	*p* Value
	(*n* = 94)	(*n* = 203)	
Age (years)	6.3 ± 4.3	6.8 ± 4.0	0.2683
Sex (%)			0.486
Male	55.3	59.6	
Female	44.7	40.4	
Seating position (%)			0.1278
Front-seat passenger	34.8	26	
Rear-seat passenger	65.2	74	
Systolic blood pressure (mmHg)	119.1 ± 21.3	115.4 ± 21.1	0.1777
Diastolic blood pressure (mmHg)	67.5 ± 17.2	70.2 ± 15.3	0.1897
Heart rate (beats/min)	116.8 ± 27.9	110.6 ± 29.0	0.1084
Respiration rate (breaths/min)	26.5 ± 10.2	25.3 ± 9.6	0.2064
Body temperature (°C)	36.4 ± 1.0	36.8 ± 0.8	<0.0001
Glasgow Coma Scale	12.4 ± 3.6	13.5 ± 2.9	0.0041
FAST positive (%)	7.5	10.3	0.4694

**Table 3 healthcare-11-01240-t003:** Comparisons of AIS by body region for the injury group with hospital arrivals from 2004 to 2007 and that from 2017 to 2019. * Median values of 2004 to 2007 injury group were significantly higher than those of the group from 2017 to 2019.

	2004 to 2007 Injury Group	2017 to 2019 Injury Group	*p* Value
	(*n* = 94)	(*n* = 203)	
AIS, median (IQR)			
Head *	1.0 (0.0–3.0)	0.0 (0.0–2.0)	0.0080
Face *	0.0 (0.0–0.0)	0.0 (0.0–0.0)	0.0193
Neck *	0.0 (0.0–1.0)	0.0 (0.0–0.0)	0.0385
Chest	0.0 (0.0–0.0)	0.0 (0.0–0.0)	0.8873
Abdomen	0.0 (0.0–0.0)	0.0 (0.0–0.0)	0.9382
Spine	0.0 (0.0–0.0)	0.0 (0.0–0.0)	0.7340
Upper extremities	0.0 (0.0–0.0)	0.0 (0.0–0.0)	0.3471
Lower extremities	0.0 (0.0–1.0)	0.0 (0.0–0.0)	0.1883

**Table 4 healthcare-11-01240-t004:** Comparisons of ISS, RTS and TRISSPS for the injury group with hospital arrivals from 2004 to 2007 and that from 2017 to 2019. ** Median values of 2004 to 2007 injury group were significantly lower than those of the group from 2017 to 2019.

	2004 to 2007 Injury Group	2017 to 2019 Injury Group	*p* Value
	(*n* = 94)	(*n* = 203)	
ISS, median (IQR)	9.0 (2.0–20.0)	9.0 (4.0–17.0)	0.6884
RTS, median (IQR) **	7.55 (6.61–7.84)	7.84 (7.55–7.84)	0.0009
TRISSPs, median (IQR)	0.99 (0.97–1.00)	0.99 (0.9–1.00)	0.5965

**Table 5 healthcare-11-01240-t005:** Comparisons of patient examinations for the injury group with hospital arrivals from 2004 to 2007 and that from 2017 to 2019.

	2004 to 2007 Injury Group	2017 to 2019 Injury Group	*p* Value
	(*n* = 94)	(*n* = 203)	
CT execution (%)	76 (81.7)	174 (86.1)	0.3268
CT (head)	69 (74.2)	145 (71.8)	0.6663
CT (neck)	23 (24.7)	112 (38.0)	<0.0001
CT (chest)	34 (36.6)	124 (61.4)	<0.0001
CT (abdomen)	43 (46.2)	122 (60.4)	0.0228
CT (pelvis)	26 (28.0)	106 (52.5)	<0.0001
CT (spine)	5 (5.38)	61 (30.2)	<0.0001
Transfusion within 24 h in admission (%)	9 (9.57)	12 (5.91)	0.0850

**Table 6 healthcare-11-01240-t006:** Comparisons of patient medical treatments for the injury group with hospital arrivals from 2004 to 2007 and that from 2017 to 2019.

	2004 to 2007 Injury Group	2017 to 2019 Injury Group	*p* Value
	(*n* = 94)	(*n* = 203)	
Angiography (%)			
Head	0 (0.00)	0 (0.00)	
Neck	0 (0.00)	0 (0.00)	
Chest	0 (0.00)	1 (0.49)	0.4955
Abdomen	1 (1.06)	7 (3.45)	0.2378
Pelvis	0 (0.00)	2 (0.99)	0.3342
Spine	0 (0.00)	0 (0.00)	
Primary surgery (%)			
Craniotomy	6 (6.38)	4 (1.97)	0.0499
Perforator	1 (1.06)	3 (1.48)	0.7734
Open thoracotomy	1 (1.06)	3 (1.48)	0.7734
Laparotomy	3 (3.19)	11 (5.42)	0.3996
Bone fracture	13 (13.8)	25 (12.3)	0.7163
Revascularization	0 (0.00)	1 (0.49)	0.4955
Hemostasis	1 (1.06)	0 (0.00)	0.1410
Limb followed by surgery	0 (0.00)	0 (0.00)	
With laparoscopy	0 (0.00)	1 (0.49)	0.4955
TAE	1 (1.06)	3 (1.48)	0.7734

## Data Availability

The data that support the findings of this study are available from Japan Trauma Care and Research but restrictions apply to the availability of these data, which were used under license for the current study, and so are not publicly available. Data are however available from the authors upon reasonable request and with permission of Japan Trauma Care and Research.

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
