# Peer review of "Increased CT Use and No Change in Injury Severity among Child Motor Vehicle Victims: A National Trauma Database Study in Japan"

_healthcare, 2023, doi:10.3390/healthcare11091240_

Round 1

Reviewer 1 Report

This is my pleasure to review this manuscript. 

I have several suggestions and comments:

1.    There are two major findings in this study. First, More CT examinations were performed in the 2017-19 era compared to 2004-07 era. Second, the severity of pediatric motor vehicle injuries was similar for these two groups. Are these two findings related? If not, may beIncreased CT use and no change in injury severity among child motor vehicle…..is a more appropriate title for this maniuscript.

2.    Second sentence of the Introduction section: Worldwide, among fatally injured children under age 19 years, 45 children required hospitalization and 1300 needed medical care…., it is confusing that only 45 children required hospitalization worldwide. Please clarify.

3.    The number of hospitals that participated in JTDB are increasing over the years. From 55 hospitals in 2005 to more than 303 hospitals in 2021. In this study, the body temperature, RTS and GCS were significantly lower in the 2004-07 group. Is it possible that because large trauma centers participated in JTDB earlier than smaller hospitals so higher percentage of severe cases were recorded in JTDB in earlier years? If so, how would it affect the current result?

4.    The AIS score of neck was higher in 2004-07, but more CT of neck was performed in 2017-19, is there any possible explanations?

5.    In the conclusion, the authors stated that Because the rate of performing CT has increased, with no change in injury severity of the trunk and extremities, limiting the number of CT examinations is recommended for child motor vehicle passengers involved in RTCs.

I am not sure if this is a logical statement. Why were the doctors in the 2017-19 era prefer to perform more CT compared to doctors in 2004-07 era? Were there any update of practicing guidelines or trend for the use of CT scan as an initial evaluation tool in pediatric trauma? Please address this issue in the discussion if possible.

Author Response

It is listed a point-by-point response to the reviewer's comment in a separate file.

Reviewer 2 Report

The authors drew attention the excess using of CT, which has come to be a significant issue and a major burden on healthcare systems in the modern medical landscape, based on children who have been exposed to traffic accidents. 

However, the authors should to explain why they decided to evaluate a 3-year study period (2017-2019) using a 4-year control period (2004-2007). 

Unnecessary use of CT is an important problem not only in pediatric patients who have been exposed to traffic accidents, but also in adult patients (Broder J, Warshauer DM. Increasing utilization of computed tomography in the adult emergency department, 2000-2005. Emerg Radiol. 2006;13(1):25-30. doi:10.1007/s10140-006-0493-9).

The authors should also take into account the potential rise in medical malpractice lawsuits over the same time period and the development of defensive medicine when questioning the rise in the unnecessary usage of CT.

Author Response

(The authors gave the same response as above.)

Reviewer 3 Report

Good work, but the paper is not suitable for publication in a Special Issue titled "Innovations in Forensic Medicine".

The paper could be rather suitable for publication in journals about forensic pathology and/or pediatrics.

Author Response

(The authors gave the same response as above.)

Round 2

Reviewer 2 Report

Necessary revisions were made by the authors, taking into account my suggestions. I think the acceptance of the article is appropriate.

Author Response

Responses to the comments of Reviewer #2 

Thank you for your helpful review of our manuscript. Your comments enabled us to substantially improve the quality of our manuscript.

Response:

Following your suggestion, we have proofread the English text.